# Combined High-Throughput Approaches Reveal the Signals Driven by Skin and Blood Environments and Define the Tumor Heterogeneity in Sézary Syndrome

**DOI:** 10.3390/cancers14122847

**Published:** 2022-06-09

**Authors:** Cristina Cristofoletti, Antonella Bresin, Martina Fioretti, Giandomenico Russo, Maria Grazia Narducci

**Affiliations:** Istituto Dermopatico dell’Immacolata, IDI-IRCCS, Via dei Monti di Creta, 104, 00167 Rome, Italy; a.bresin@idi.it (A.B.); m.fioretti@idi.it (M.F.); g.russo@idi.it (G.R.)

**Keywords:** cutaneous T-cell lymphoma, Sézary syndrome, blood and skin microenvironment, single cell analysis, transcriptome, biochemical signals, immunophenotype, heterogeneity

## Abstract

**Simple Summary:**

Sézary syndrome (SS) is a leukemic and incurable variant of cutaneous T-cell lymphoma characterized by the accumulation of neoplastic CD4+ lymphocytes in the blood, lymph nodes, and skin. With the exception of allogenic transplantation, no curative chance is available to treat SS, and it is a priority to find new therapies that target SS cells within all disease compartments. This review aims to summarize the more recent analyses conducted on skin- and blood-derived SS cells concurrently obtained from the same SS patients. The results highlighted that skin-SS cells were more active/proliferating with respect to matched blood SS cells that instead appeared quiescent. These data shed the light on the possibility to treat blood and skin SS cells with different compounds, respectively. Moreover, this review recaps the more recent findings on the heterogeneity of circulating SS cells that presented a series of novel markers that could improve diagnosis, prognosis and therapy of this lymphoma.

**Abstract:**

Sézary syndrome (SS) is an aggressive variant of cutaneous t-cell lymphoma characterized by the accumulation of neoplastic CD4+ lymphocytes—the SS cells—mainly in blood, lymph nodes, and skin. The tumor spread pattern of SS makes this lymphoma a unique model of disease that allows a concurrent blood and skin sampling for analysis. This review summarizes the recent studies highlighting the transcriptional programs triggered by the crosstalk between SS cells and blood–skin microenvironments. Emerging data proved that skin-derived SS cells show consistently higher activation/proliferation rates, mainly driven by T-cell receptor signaling with respect to matched blood SS cells that instead appear quiescent. Biochemical analyses also demonstrated an hyperactivation of PI3K/AKT/mTOR, a targetable pathway by multiple inhibitors currently in clinical trials, in skin SS cells compared with a paired blood counterpart. These results indicated that active and quiescent SS cells coexist in this lymphoma, and that they could be respectively treated with different therapeutics. Finally, this review underlines the more recent discoveries into the heterogeneity of circulating SS cells, highlighting a series of novel markers that could improve the diagnosis and that represent novel therapeutic targets (GPR15, PTPN13, KLRB1, and ITGB1) as well as new genetic markers (PD-1 and CD39) able to stratify SS patients for disease aggressiveness.

## 1. Introduction

Cutaneous T-cell lymphomas (CTCLs) include a large spectrum of mature T-cell neoplasms characterized by the accumulation of neoplastic CD4+ T lymphocytes in the skin.

The most common subtype of CTCL is mycosis fungoides (MF), which represents 60% of CTCL cases, while the much rarer variant is Sézary syndrome (SS), which accounts for approximately 5% [1]. The majority of patients with an early MF show a skin-restricted infiltration of malignant cells and an indolent course [1], but about 15% of them can progress toward advanced stages characterized by tumors and erythroderma and that show a decreased survival at less than 5 years [2,3]. On the contrary, SS is a rare and leukemic subtype of CTCL showing, ab initio, the simultaneous presence of neoplastic lymphocytes mainly in the blood, lymph nodes, and skin.

The relationship between MF and SS is still debated. For a long time, SS was considered an aggressive form of MF, but growing data from genome analyses have demonstrated specific chromosomal alterations occurring in MF but not in SS, and vice versa, suggesting that they are two separate clinical entities [4]. These findings are also reinforced by the different transcriptomic profiles and cell surface markers expressed by MF and SS. According to these data, Campbell et al. indicated that MFs arise from skin resident effector memory (EM) T cells, whereas SS arises from central memory (CM) T-lymphocytes [5]. However, the cell of origin of SS and MF is still undetermined. 

Despite the great efforts to characterize SS pathogenesis, it remains an incurable disease, with 5-year overall survival (OS) rates of 15% to 40% for stage IVA and of 0% to 15% for stage IVB [2,6]. To date, most studies have explored the intrinsic molecular features of SS, demonstrating that it is characterized by a complex profile of chromosome aberrations [7,8,9] and a broad range of genes variously affected by somatic copy-number alterations and somatic single-nucleotide variants that are involved predominantly in T-cell activation and apoptosis, activation of NF-kB, JAK/STAT signaling, chromatin remodeling, and DNA damage response [10,11].

More recently, attention has shifted toward extrinsic mechanisms that impact the pathophysiology of cancer. Indeed, understanding how the tumor microenvironment sustains neoplastic cells has become focal in cancer research [12]. Regarding CTCL, many reports have already described the complex cellular interactions between malignant lymphocytes and skin elements that play a key role in CTCL pathogenesis [13,14,15,16,17].

Unlike metastatic spread, cutaneous lymphoma dissemination does not reflect tumor progression, but rather a conserved physiological behavior of neoplastic lymphocytes [18]. This characteristic indicates the strong dependence of these cells on the cutaneous niche that supports their proliferation and survival through the release of nutrients and the induction of cellular signals [19,20,21,22,23]. Tumor is a dynamic disease, and during its progression usually acquires a greater heterogeneity. This implies that the original cancer cells acquire different molecular signatures, generating an intratumor heterogeneity. Such a condition is the greatest cause of failure in cancer therapies because the spatial and temporal genetic and epigenetic diversity and/or plastic gene expression in cancer cells is often associated with the mechanism of drug resistance [24]. A strong contribution to heterogeneity is also made by the tumor microenvironment, which triggers distinct and specific signals to tumor cells. This last aspect assumes a particular importance in CTCL, given the SS cells simultaneously invade the bloodstream and the skin district, where they interact with keratinocytes, as well as bystander stromal and immune cells [25]. The impact of fluid and solid environment on tumor cells adds complexity to the disease and produces intratumor heterogeneity.

To provide a new perspective compared with the latest reviews on molecular heterogeneity and deregulated pathways found in CTCLs [26,27], here we focused on comparison analyses performed between paired skin and blood samples of SS cells. Emerging data describe the molecular signals triggered in these two SS cell subpopulations. Special emphasis was also given to new insights into the transcriptional heterogeneity found in blood-derived SS cells, highlighting new markers that could improve the diagnosis and prognosis of this aggressive form of CTCL.

## 2. Blood and Skin Microenvironments Drive the Transcriptional Program and Signaling of SS Cells

A role for the skin microenvironment in SS pathogenesis has long been hypothesized, but its function in vivo is poorly known. Lately, some investigations have analyzed matched blood- and skin-derived SS cells, revealing substantial differences between these two cell subsets. In this section, we discuss the main results obtained so far (Figure 1).

Roelens et al. were the first authors who studied this aspect [28]. In a study of 16 SS patients, they observed that compared to blood SS cells, matched skin SS cells showed a more activated phenotype with a deregulated expression of interleukin receptors such as CD25 (IL-2R), CD215 (IL-15R), and CD127 (IL-7R), suggesting that skin SS cells have a major proliferative and survival advantage in response to specific interleukins involved in SS pathogenesis. These authors demonstrated that skin SS cells also displayed a higher expression of chemokine receptors such as CXCR3, CCR6, and CCR10, whereas CCR4 resulted in downregulation, a finding that could explain the lower efficacy at the skin level of mogamulizumab, a therapeutic anti-CCR4 monoclonal antibody [29].

The recent development of single-cell approaches has allowed the analysis of the transcriptomes, cell surface markers, and genomes of individual cancer cells with unprecedented resolution. For example, Herrera et al. [30] recently employed a multimodal approach that allowed the detection of the T-cell receptor (TCR) clonality (TCR β CD3), single-cell transcriptome, and surface protein expression, as well as genetic analyses based on copy-number variations (CNVs) on matched skin and blood samples derived from five SS patients [30]. The authors proved that matched skin- and blood-derived SS cells shared the same TCR clonotypes, demonstrating that the same neoplastic lymphocytes could be distributed in both tumor environments. Transcriptional analyses highlighted that blood SS cells contained several clusters characterized by different expression profiles (high transcriptional heterogeneity), whereas skin-derived SS cells mainly clustered together, revealing a more homogeneous expression profile (low transcriptional heterogeneity) (Figure 2). Phylogenetic analyses suggested a continuous migration of SS cells between the skin and blood, rather than a monodirectional migration predicted for a tissue of origin of the disease [30].

Thus, the high and low transcriptional heterogeneity found in blood- and skin-derived SS cells may be caused by a reprogramming of SS cells due to their passage from one tumor environment to another. For example, from blood, where they overcome hemodynamic forces and interact with blood cells, to skin, where they interact with stromal and immune cells, or vice versa. By performing a transcriptome comparison, the authors also proved that skin SS cells were associated with a more activated phenotype, with a strong upregulation of several transcription factors induced by T-cell activation, TCR ligation, and mitogens, including regulators of the cell cycle (Figure 1). Among these, the activated status of skin SS cells was associated with a consistent upregulation of PD-1 and with a greater proliferative capacity than that observed in paired circulating cells, as also demonstrated by Cristofoletti et al. [31]. Conversely, the quiescent status of blood SS cells was linked to KLF2, TCF7, and CD62L overexpression, consistent with their role in T-cell resting [32,33,34] (Figure 1). It should be noted that PD-1 is an inhibitory receptor that is upregulated following T-cell activation, with the aim of avoiding the damage of cell hyperactivation, and thus assuming an exhaustion role [35]. Therefore, the PD-1 overexpression found in skin SS cells compared with paired blood SS cells seems to reflect their activation status [30].

Overall, these experiments conducted on paired skin–blood SS cells revealed which molecular drivers are requested to arrange the tissue distribution of SS cells. They also indicate the possibility of using drugs, administrated simultaneously or in sequence, that are able to interfere with resting blood SS cells and proliferating skin SS counterparts.

## 3. Skin SS Cells Exhibit a Hyperactivated PI3K/AKT/mTOR Signaling Compared to the Matched Blood Counterpart: Spotlight on Skin Microenvironment

Activation of CD4+ T cells upon engagement of the TCR and costimulatory receptors lead to changes in expression profiles, remodeling of the T-cell proteome, and differentiation into effector CD4+ T-cell subpopulations [36]. TCR pathway alterations are frequently observed in SS [7,9,37,38,39,40]. Engagement of TCR activates, among others, the PI3K/AKT/mTOR pathway, leading to T-cell proliferation, survival, and differentiation, as well as cytokine production [41].

Throughout the years, many investigations proved that PI3K/AKT/mTOR signaling is strongly involved in SS pathogenesis. Early genomic studies demonstrated that PTEN, the major antagonist of PI3K/AKT signaling [42], is frequently deleted at the monoallelic level in both MF [43] and SS, in which PTEN is also downregulated by miR-21, miR-106b, and miR-486 [44]. More recently, our study investigated the CNVs of members belonging to this cascade in 43 SS patients highlighting recurrent alterations; namely, a loss in tumor suppressors such as LKB1 (48%), PTEN (39%), and PDCD4 (35%), and a gain in the proto-oncogene P70S6K (30%). Each of these CNVs, whether evaluated individually or in combination, was associated with the reduced survival of SS patients [31]. It was notable that these genetic alterations were differently distributed among patients, thus contributing to the interpatient heterogeneity.

According to these results, the therapeutic efficacy of PI3K/mTOR inhibitors was found to be variable among patients and correlated with their genomic status, thus confirming the strong impact of heterogeneity in the clinical course of SS patients as well [45].

Functional experiments demonstrated that many cytokines and growth factors detected in SS skin lesions are able to activate PI3K/AKT/mTOR signaling. The first evidence came from Marzec M. et al. [46], who demonstrated that IL-2 triggered this pathway in activated primary SS cells and that its inhibition, obtained via treatment with the mTORC1 inhibitor rapamycin [46,47], was able to block SS cell growth in vitro [46] as well in a xenograft T-cell lymphoma mouse model [48].

Other investigations have been performed in this direction: Murga-Zamalloa et al. [49] studied the interaction between CTCL cells and lymphoma-associated macrophages known to play a critical role in disease progression, and demonstrated that autocrine colony stimulating factor 1 (CSF1) activated AKT/mTOR signaling and promoted T-cell lymphoma viability. Activation occurred upon binding to the CSF1 receptor, which is highly expressed by both macrophages and lymphoma cells.

It is also interesting that the IL-31/IL-31 receptor axis involved in the mechanism of itch, which is one of the worst symptoms of SS [50], is capable of driving PI3K/AKT signaling in a plethora of skin diseases [51].

Immunohistochemistry confirmed that this pathway is activated in skin-infiltrating SS cells by the detection of high levels of phosphorylated forms of AKT, mTOR, P70S6K, S6RP, and 4EBP1 [46,52]. We further deepened this aspect [31] by demonstrating a higher phosphorylation level of these proteins, particularly of mTOR, as well as a greater proliferation index in skin-derived SS cells when compared to blood-derived SS cells concurrently obtained from the same patients (Figure 1). We also proved that SDF-1 and CCL21 chemokines, both overexpressed in SS tissues [53,54,55,56], induced mTORC1 signaling activation, cell proliferation, and Ki67 upregulation in primary-SS cells and in a SS-derived cell line.

The skin–blood comparison approach therefore demonstrates how the skin upregulates PI3K/AKT/mTORC1 signaling, and indicates a strategy for discovering new biochemical signals that support the growth and expansion of potentially compound target SS cells (Figure 1).

## 4. Recent Advances in Intratumor Heterogeneity in Circulating SS Cells

High-throughput technologies such as RNA sequencing (RNA-seq) represented a turning point for in-depth exploration of tumor heterogeneity. There are essentially two methods of RNA-seq: bulk RNA-seq and single-cell RNA-seq (scRNA-seq). Bulk RNA-seq provides for each transcript, an average expression level in the sample, which may comprise different cell types [57]. Conversely, scRNA-seq is able to detect genes expressed even in small subclones within bulk RNA-seq. Thus, it provides a higher resolution of cellular differences and a better understanding of the function of an individual cell in the context of its microenvironment.

Recently, several different high-throughput techniques such as flow cytometry [28,58,59], TCR sequencing [60,61], TCR clonality, and surface protein expression have been used in combination with RNA-seq or scRNA-seq analyses to gain further insights into cell heterogeneity and the dysregulated gene expression emerging from SS cells.

The first of these new studies, which combined phenotypic characterization with RNA expression, was conducted by Roelens M. et al. [28]. Using an eight-color flow cytometry, the authors characterized circulating SS cells from 45 patients using TCR Vb+CD158k+ or CD158k+ (KIR3DL2) staining in combination with other specific T-cell markers. They observed that clonal SS cells were not uniformly associated with a CM phenotype, but they included minor fractions of naïve (TN), transitional memory (TTM), effector memory (EM), and stem-cell memory (TSCM) clonal subpopulations. This last subpopulation was characterized by FAS receptor expression (CD95) that influenced the cellular apoptotic resistance [62,63], as reported in adult T-cell leukemia [64]. This investigation highlighted for the first time the great diversity of clonal SS cells, shedding light on the concept of phenotypic SS cell heterogeneity (Figure 2).

In an attempt to find a specific SS cell expression signature, the authors then compared the transcriptional profiles of TN, TCM, and TSCM SS cells derived from three patients with healthy cell counterparts derived from three healthy donors. The results revealed a common signature among all SS cell subpopulations, represented mainly by the overexpression of CD158k, T-plastin, and Twist. These results could have been partly influenced by the positive selection of malignant lymphocytes based on CD158k staining, a marker that does not appear to be exclusive to SS cells, as observed in more recent investigations [65,66].

In another study, Borcherding et al. [60] combined scRNA-seq and T-cell receptor sequencing analyses in matched isolated SS cells and normal CD4+ cells derived from one SS patient. Based on mRNA expression, the authors distinguished six clusters within normal CD4+ cells that appeared to be mainly associated with a naïve phenotype, and five clusters within SS cells that contained a phenotype consistent with CM-T cells. Each cluster found was defined by the expression of the top 5–7 genes. To identify novel markers and/or therapeutic targets of SS, the authors compared the transcriptome of paired malignant and normal cells by highlighting a series of differential expressed genes. Among these, they found two novel markers for SS; namely, SAMSN1, previously reported as involved in resistance to IFNa, a key drug in CTCL treatment [67]; and TSPAN2, which is involved in the migration of lung cancer [68]. The five clusters emerging from the malignant cells showed different expression profiles, allowing the authors to trace the SS cells’ trajectory. Using this analysis, the SS cells appeared to evolve from FOXP3+ to GATA3+ or IKZF2+ (Helios) cells, possibly in response to environmental factors, such as *Staphylococcus aureus* infection [69]. Using an artificial-intelligence-based algorithm, the expression of these genes was also investigated in a large dataset of CTCL patients, demonstrating that the most predictor of CTCL early stage was FOXP3, the principal transcription factor for the Treg T-cell phenotype. These data provide new elements of the still-debated phenotype similarity between SS cells and Treg cells, and suggest that SS cell plasticity from Treg to a CM phenotype could be addressed by microenvironment signaling(s) (Figure 2) [60,70,71].

Overall, these results seemed to outline a change in the transcriptional profile during SS disease progression, and suggested a hypothesis that SS patients can be stratified for prognosis according to the percentage of Treg-like SS cells found within their clonal cells [70,72].

Najidh et al. [59] also performed an RNA-seq analysis on purified subsets of SS cells (mainly represented by CM and naïve phenotypes) and normal CD4+ T cells obtained from the same three patients and purified using a FACS cell sorting. By comparing SS and normal CD4+, the authors identified a specific common signature for SS cells. Their gene ontology analysis highlighted downregulated genes involved in an adaptative immune response, while many genes implicated in cell adhesion, migration, and signaling appeared to be upregulated. Among these, the authors found a strong and significant downregulation of THEMIS and LAIR1, which are involved in the positive selection of thymocytes. Interestingly, these two genes were also implicated in adult T-cell leukemia, which suggests a shared pathogenetic mechanism with this leukemic variant.

In an attempt to find a common cell signature among SS cell subpopulations, we performed a Venn diagram analysis using three lists of differentially expressed genes between SS cell subsets and healthy controls that emerged from the studies presented above [28,59,60] (Figure 3).

We highlighted nine shared genes. Among these, CCR4, DPP4 (CD26), TOX, PSL3, and KIR3DL2 were already widely described in SS [73,74]. The remaining four genes are less known, and all resulted in upregulation in SS; they were: GPR15, a G protein-coupled receptor that has been implicated in T-cell trafficking and chronic inflammation [75,76]; PTPN13: a tyrosine phosphatase family protein involved in cell growth and differentiation, the mitotic cycle, and oncogenic transformation [77]; KLRB1 (CD161), a killer cell lectin-like receptor that represents a promising immune checkpoint receptor [78] involved in T-cell proliferation and inhibition of natural killer cell cytotoxicity; ITGB1: a receptor implicated in proliferation, invasion, angiogenesis, immune response, and therapeutic resistance [79], which are of importance in immunotherapeutic strategies [80].

Overall, these novel genes might be useful as diagnostic factors for SS, and have relevant therapeutic implications in curing this lymphoma.

## 5. Intratumor Heterogeneity Influences the Drug Resistance in SS: A Possible Tool to Orient the Therapies In Vivo

As already mentioned, intratumor heterogeneity is an important factor in resistance or sensitivity to therapies. This aspect was recently analyzed by Buus et al. [58] in 11 samples from SS patients. Malignant cells were identified by their expression of a dominant TCR Vb rearrangement, and further characterized through 240 surface markers detected using multicolor flow cytometry. The results revealed that the SS cells encompassed different subpopulations within the same sample, exhibiting a distinct combination of surface markers mainly associated with a T-CM, naïve, and stem-cell memory T-cell phenotype (Figure 2). SS subpopulations derived from different patients were then analyzed for expression of 110 T-cell-related genes, also revealing a high heterogeneity across patients. Conversely, a cluster of five genes were highly expressed by most malignant cells. These included the two novel cancer-related genes S100A4 and S100A10, both with various functions in tumor development and drug resistance, and the three genes IL7R, CCR7, and CXCR4 involved in tumor growth and migration pathways. The authors then focused on the relationship between malignant heterogeneity and drug resistance by treating PBMCs from six patients with two histone deacetylase inhibitors (HDACi); namely, vorinostat and romidepsin, both approved by the Food and Drug Administration for refractory or relapsed CTCL [81,82]. The changes in the malignant subpopulations were then followed in vitro. Authors observed that malignant cells from all patients responded to treatments in a dose-dependent manner. However, the effects of HDACi exposure varied strongly among the patients analyzed. Indeed, although most malignant cells were eliminated by the drugs, some cell subpopulations remained unaffected by treatment. This approach could be useful for evaluating therapeutic efficacy and drug resistance in patients during therapy. To evaluate this aspect in depth, the drug sensitivity of PBMCs derived from SS patients under specific therapies should be assessed in vivo.

Very recently, Poglio et al. [83] addressed the aspect of tumor heterogeneity using long-term cell culture experiments and patient derived-xenograft (PDX) animal models. They cultured circulating SS cells derived from four patients under six defined culture conditions, and observed a large phenotypic heterogeneity within the clonal SS cells derived from each patient. They also noted that each SS sample only grew under specific culture conditions regardless of the original phenotype of SS subpopulations, suggesting that each sample could be related to different cells of origin or levels of maturation. These data further underlined the concept of interpatient heterogeneity and showed the critical role of the microenvironment in the growth efficiency of SS cells.

The authors also engrafted the SS cells into immunodeficient NOD scid gamma (NSG) mice. Using cell cultures and PDX models, the authors obtained novel SS cell lines that better represented each patient’s cell heterogeneity, despite it being reduced compared to the original SS cell composition (Figure 2). This heterogeneity reduction seemed to be generated by a mechanism of selection, rather than phenotypic plasticity.

Interestingly, preclinical tests using different drugs on these SS cell lines confirmed the critical role of heterogeneity in the therapeutic efficacy as well, as described by Buus et al. [58]. This study is of particular relevance because it may overcome the low efficiency of long-term SS cell culturing and the substantial lack of a standardized animal model to test the therapeutic efficacy in SS patients [84].

## 6. Beyond Genetic Interpatient Heterogeneity: What Is Next?

In the last decade, a consistent number of studies identified a heterogeneous complex scenario of genomic alterations and/or focal deletions of importance in the diagnosis and prognosis of SS [7,37,74,85]. These studies revealed a high interpatient heterogeneity demonstrated by a plethora of different chromosomal alterations associated with each patient. Among these, the most common were copy-number variations in chromosome 8q and isochromosomes 17q, 10q, 2p, 11q, and 9p [11,86]. At the gene level, the main drivers were TP53, RB1, PTEN, DNMT3A, ARID1A, CDKN2A, CDKN1B, and ZEB1 tumor-suppressor genes, all of which were lost or inactivated [9,11]. Genomic diversity in SS also involves multiple pathways, mainly including the T-cell receptor, NF-kB, JAK/STAT signaling, apoptosis, chromatin remodeling, and DNA damage response [11,37,87].

The integration of genomic data with functional analyses is highlighting novel and specific genetic markers able to stratify SS patients for therapeutic treatments and prognosis. Using this approach, Park et al. [61] demonstrated that the genomic status of PD-1 determined the level of SS cell exhaustion, and stratified the patients based on their SS cell proliferation ability. After stimulating the SS cells with TCR agonists, these authors observed that some samples, which were defined as “fully exhausted”, did not proliferate at all. They showed an increased expression of inhibitory receptors such as PD-1 and TIGIT, as well as CD39 [88,89], and produced the CXCL13 chemokine, but not IL-2.

In the second group, which was defined as “moderately exhausted”, the samples showed a modest proliferative response, and produced CXCL13 and IL-2 and expressed PD-1, TIGIT, CTLA4, and LAG3, but not CD39.

In contrast, “not-exhausted” SS cells were characterized by a high proliferative ability and by production of consistent levels of effector cytokines. These samples displayed a low PD-1 and TIGIT expression, harbored the PD-1 gene deletion, and showed an up-regulated cell-cycle transcriptional signature driven by the FOXM1 transcription factor. Using a mouse model, the authors proved that PD-1 deletion was sufficient to reverse the exhaustion phenotype [61,90]. Genetic analyses conducted on a greater number of SS samples demonstrated that PD-1 loss was more frequently found in highly proliferating samples (not-exhausted), and that this deletion correlated with a worse clinical feature and poor overall survival.

This study suggested that PD-1 genomic status could help stratify CTCL patients based on disease aggressiveness and become a useful marker for selecting patients for anti-PD-1/PD-L1 immunotherapies. Since these therapeutic approaches are effective for a discrete percentage of CTCL patients (overall response rate of 38%) as demonstrated in a phase II study [91], the evaluation of PD-1 status might be useful in selecting the responder patients before clinical enrollment (ClinicalTrials.gov: NCT03385226, NCT04118868).

Another example of a genetic marker of interpatient heterogeneity is CD39, which, as mentioned above, was found to be coexpressed with PD-1 in the “fully exhausted” SS samples. CD39 is an ectonucleotidase that is responsible, together with CD73, for a cascade of events to convert proinflammatory ATP into immunosuppressive adenosine in the tumor microenvironment [92]. Interestingly, CD39 expression is genetically controlled by the SNP rs10748643 A/G within the promoter region of the CD39 gene [93], as demonstrated in Treg cells [94], peripheral activated CD4+ T cells [95], tumor-infiltrating CD8+ T cells [96], and SS cells, as we recently established. In this study, we observed that individuals carrying the A/G-G/G genotype showed a significantly higher frequency of clonal CD4+CD39+ SS cells (CD39^high^) compared with A/A patients (CD39^low^). Conversely to what was observed in solid tumors, we also noted that high CD39 expression correlated with a better prognosis, a finding that, in our cohort of 47 SS patients, appeared to be justified by the increased apoptotic susceptibility and reduced production of IL-2 found in the CD39^high^ SS cells when compared with the CD39^low^ SS cells [97]. These results indicated that PD-1 and CD39 represent novel genetic markers that can orient the therapeutic choices for stratified patients.

## 7. Conclusions

SS represents an ideal model for understanding the effects of blood and skin microenvironments on lymphoma cells. The recent combination of high-throughput analyses has allowed the investigation of skin- and blood-derived SS cells concurrently obtained from the same SS patients. Emerging data proved that skin-derived SS cells showed consistent higher activation/proliferation rates, mainly driven by TCR signaling, as well as a more homogeneous transcriptional profile with respect to blood SS cells, which appeared quiescent. This comparative approach provided a base for understanding the molecular drivers requested to arrange the skin–blood distribution of SS cells; it also shed light on the potential use of drugs, administrated simultaneously or in sequence, that are able to interfere with resting blood SS cells and proliferating skin SS counterparts.

A biochemical comparative approach performed using paired skin–blood SS cells also demonstrated an hyperactivation at the skin level of PI3K/AKT/mTOR, a pathway that can be targeted by multiple inhibitors currently in clinical trials. So, this method also might be useful in addressing targeted therapy and in discovering novel biochemical signals that support the growth and expansion of SS cells that can be targeted by specific treatments.

RNA-seq and scRNA-seq analyses combined with high-throughput techniques to investigate TCR clonality and genetic and surface protein expression have demonstrated a consistent intratumor heterogeneity, as highlighted by distinct blood SS cell subpopulations found within the bulk SS cells derived from single patients. These subpopulations were mainly represented by CM, naïve, EM, and stem CM clonal subsets. Each of them was associated with a specific transcriptional profile, although a cluster of four novel common genes—GPR15, PTPN13, CD161, and ITGB1—resulted in downregulation when compared to healthy controls, as emerged in our Venn diagram. These markers might represent novel genes that are useful as diagnostic factors for SS, with relevant therapeutic implications.

The SS cell subpopulations characterized by specific transcriptional and phenotypical profiles are also particularly useful in the identification of drug-resistant cell sub-sets that contribute to therapy failure. Thus, the assessment of drug sensitivity of PBMCs derived from SS patients under specific therapies might help to evaluate therapeutic responses or sensitivity in vivo.

Intratumor heterogeneity of blood-derived SS cells can be investigated by culturing patients’ SS cells for a prolonged time (long-term cell culture) or by transplanting them into NSG mice. The emerging SS cell lines better represent the original SS cell heterogeneity, and they can be used to test drug efficacy. Therefore, this approach can potentially overcome the lack of an SS animal model for preclinical studies.

Finally, despite the high interpatient heterogeneity largely described for SS, new explorations have discovered that PD-1 genomic status, as well as gene-driven CD39 expression (SNP_rs10748643), represent two novel genetic markers that are able to stratify SS patients by disease aggressiveness and potentially guide such patients to therapies.

## 8. Summary

Compared with paired blood SS cells, skin SS cells showed: (1) a higher activation/proliferation rate mainly driven by TCR signaling; and (2) hyperactivation of PI3K/AKT/mTOR, a pathway targetable by multiple inhibitors currently in clinical trials.

The new insights regarding the transcriptomic heterogeneity found in blood SS cells underline novel markers commonly expressed by clonal SS subsets that can improve the diagnosis and prognosis of and therapy for this lymphoma.

## Figures and Tables

**Figure 1 cancers-14-02847-f001:**
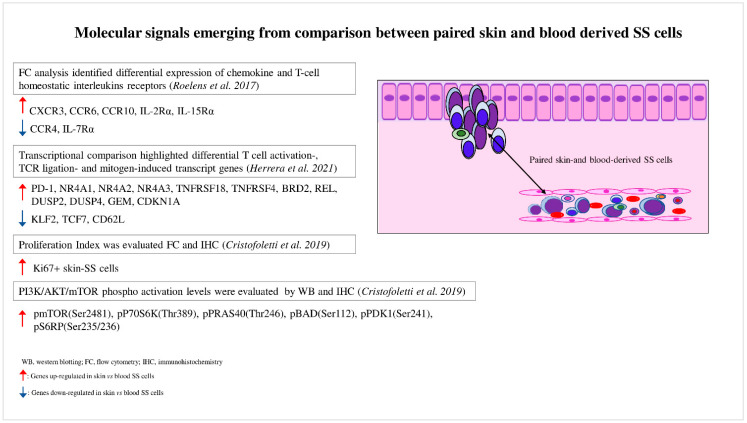
Molecular signals emerging from comparison between paired skin- and blood-derived SS cells. Comparison analyses performed between paired skin- and blood-derived SS cells highlight the molecular signals triggered in these two SS cell subpopulations.

**Figure 2 cancers-14-02847-f002:**
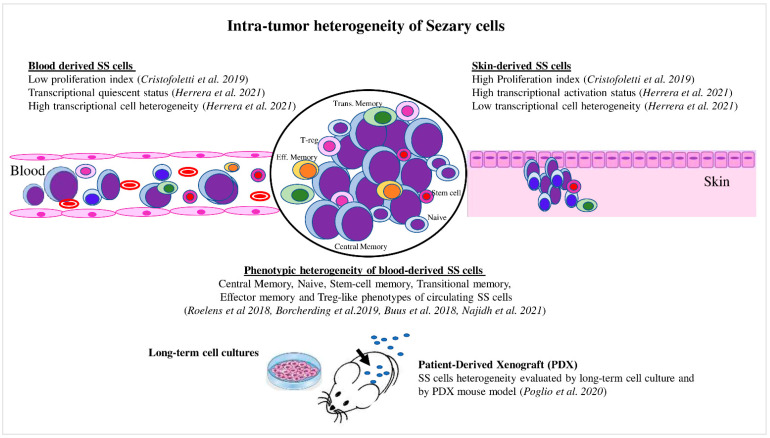
Intratumor heterogeneity of Sézary cells. Skin- and blood-derived SS cells showed differences in proliferation capacity, expression of activation markers, and level of transcriptional heterogeneity. Blood SS cells showed phenotypic heterogeneity, which also was evaluated under specific culture conditions and in the PDX mouse model.

**Figure 3 cancers-14-02847-f003:**
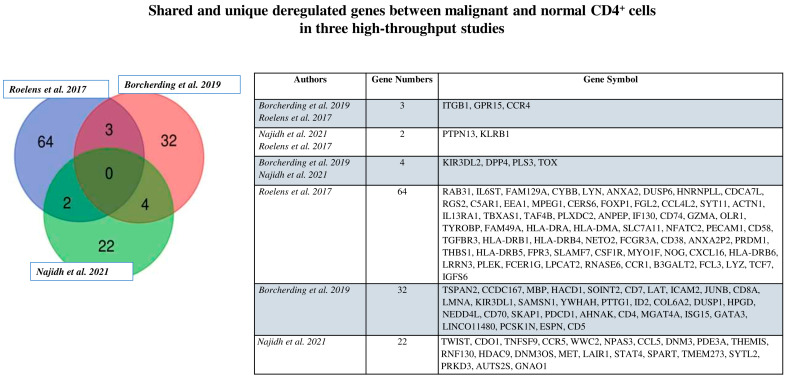
Shared and unique deregulated genes between malignant and normal CD4+ cells in three high-throughput studies.

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
