# Peer review of "Combined High-Throughput Approaches Reveal the Signals Driven by Skin and Blood Environments and Define the Tumor Heterogeneity in Sézary Syndrome"

_cancers, 2022, doi:10.3390/cancers14122847_

Round 1

Reviewer 1 Report

I thank to the editors for the opportunity to review this work, beside I would also like to congratulate the authors for the made effort. This review aims to summarize the more recent analyses conducted on skin and blood-derived Sezary Syndrome (SS) cells concurrently obtained from the same SS patients, highlighting the transcriptional programs triggered by the crosstalk between SS cells and blood-skin microenvironments. In addition, the novelty of this review is that includes the more recent discoveries into the heterogeneity of circulating SS cells, particularly useful for the evaluation of therapeutic responses or sensitivity in vivo. Nevertheless, and in spite of the significant amount of work performed, some important issues have to be consider.

General comments:

  1. Are others reviews there similar to your study? For example, after a quick search, I have found the following reviews, precisely in this same journal:
  • Rassek K, Iżykowska K. Single-Cell Heterogeneity of Cutaneous T-Cell Lymphomas Revealed Using RNA-Seq Technologies. Cancers (Basel). 2020 Jul 31;12(8):2129. doi: 10.3390/cancers12082129. PMID: 32751918; PMCID: PMC7464763.
  • García-Díaz N, Piris MÁ, Ortiz-Romero PL, Vaqué JP. Mycosis Fungoides and Sézary Syndrome: An Integrative Review of the Pathophysiology, Molecular Drivers, and Targeted Therapy. Cancers (Basel). 2021 Apr 16;13(8):1931. doi: 10.3390/cancers13081931. PMID: 33923722; PMCID: PMC8074086.

What does relevant information provide your review that these reviews do not offer?

  1. In my opinion, Figure 1 and 2 should be much clearer. The authors could express the same concepts in a clearer and more graphically represented way. In the same way, they should add legends to better explain the figures.
  2. The conclusions show the most important results that the review aims to highlight, but are too long. Authors should summarise in a final sentence the most salient finding of the review.

Minor comments:

  1. The authors should add the meaning of the abbreviations: IVA and IVB (P2L57); HD (P6L221).
  2. In Figure 2, what is the meaning of “Hyperactivation of PI3K/Akt/mTOR Assay”?

Author Response

Point-by-point response 

We really thank the reviewers for their comments which were very helpful in improving the strength of our manuscript. We listed below the point-by-point response to each of the Reviewers’ observations in blue, as well as listed text revised changes in red. 

Reviewer n. 1                Comments and Suggestions for Authors 

I thank to the editors for the opportunity to review this work, beside I would also like to congratulate the authors for the made effort. This review aims to summarize the more recent analyses conducted on skin and blood-derived Sezary Syndrome (SS) cells concurrently obtained from the same SS patients, highlighting the transcriptional programs triggered by the crosstalk between SS cells and blood-skin microenvironments. In addition, the novelty of this review is that includes the more recent discoveries into the heterogeneity of circulating SS cells, particularly useful for the evaluation of therapeutic responses or sensitivity in vivo. Nevertheless, and in spite of the significant amount of work performed, some important issues have to be consider. 

General comments

1 - Are others reviews there similar to your study? For example, after a quick search, I have found the following reviews, precisely in this same journal:

Rassek K, Iżykowska K. Single-Cell Heterogeneity of Cutaneous T-Cell Lymphomas Revealed Using RNA-Seq Technologies. Cancers (Basel). 2020 Jul 31;12(8):2129. doi: 10.3390/cancers12082129. PMID: 32751918; PMCID: PMC7464763. 

García-Díaz N, Piris MÁ, Ortiz-Romero PL, Vaqué JP. Mycosis Fungoides and Sézary Syndrome: An Integrative Review of the Pathophysiology, Molecular Drivers, and Targeted Therapy. Cancers (Basel). 2021 Apr 16;13(8):1931. doi: 10.3390/cancers13081931. PMID: 33923722; PMCID: PMC8074086.

What does relevant information provide your review that these reviews do not offer? 

We thank the Reviewer for their careful reading of the manuscript and their constructive remarks. Please find our answer to comments in blue as well as suggested text changes in red. As requested, we highlighted the differences between the previous reviews and this one (P2 L88-94). The two reviews indicated are now included in the references 26, 27.  

2 - In my opinion, Figure 1 and 2 should be much clearer. The authors could express the same concepts in a clearer and more graphically represented way. In the same way, they should add legends to better explain the figures. 

Following the reviewer's indication, we have modified Figure 1 and 2.  Accordingly, we have better explained both figures in their respective legends

3 - The conclusions show the most important results that the review aims to highlight but are too long. Authors should summarize in a final sentence the most salient finding of the review. 

As requested, we summarized the most salient finding under a new paragraph titled “8. Summary” (P10 L426-432). 

  Minor comments: 

1 - The authors should add the meaning of the abbreviations: IVA and IVB (P2L57); HD (P6L221). 

IVA and IVB stages for MF/SS are defined by ISCL/EORTC. For clarity, we have now added a more appropriate reference to the relative sentence (P2 L61) and to the bibliography, [6] (Olsen E. et al Blood 2007). We replaced HD with “healthy donors” (P6 L229). 

2 - In Figure 2, what is the meaning of ““Hyperactivation of PI3K/Akt/mTOR Assay” in Figure 2. 

We removed the sentence regarding “Hyperactivation of PI3K/Akt/mTOR Assay” in Figure 2 because this concept is already described in Figure 1. In a similar manner we also removed the sentences regarding the deregulation of cytokine (IL7,2,15) and chemokines (CCR4 and CCR6) also described in Figure.

Reviewer 2 Report

I think that this is a well-written and very extensive review on genetic studies on Sezary syndrome patients. I had liked the conclusions and the study analysis.

Author Response

Reviewer n. 2               Comments and Suggestions for Authors

I think that this is a well-written and very extensive review on genetic studies on Sezary syndrome patients. I had liked the conclusions and the study analysis.

We thank the Reviewer for the positive comments.

Reviewer 3 Report

Excellent review, very nicely written. The overlap between the 3 RNA-Seq studies is quite telling. Overall figures are summarizing nicely. Only a few minor issues:

1 - Progression rate from early MF is too high from most literature. More ~15%

2 - Two major types of CTCL are MF and CD30+

3 - A bit more time should be spent on the use of anti-PD-1/PD-L1 therapy for MF and for SS.

Author Response

Reviewer n. 3                Comments and Suggestions for Authors 

Excellent review, very nicely written. The overlap between the 3 RNA-Seq studies is quite telling. Overall figures are summarizing nicely. Only a few minor issues: 

1 - Progression rate from early MF is too high from most literature. More ~15% 

We thank the Reviewer for the positive comment and careful review. Please find our answer to comments in blue as well as suggested text changes in red. 

We have corrected the percentage (P2 L46) 

2 - Two major types of CTCL are MF and CD30+ 

We changed the sentence (P1 L43- P2 L45) 

3 - A bit more time should be spent on the use of anti-PD-1/PD-L1 therapy for MF and for SS. 

As suggested by the reviewer we extend the discussion on the usage of anti-PD-1/PD-L1 therapy for MF and for SS (P9 L365-368).  

Round 2

Reviewer 1 Report

Comments to the Author:

I commend the effort made by the authors to revise this manuscript. The authors have answered correctly all my questions and they have modified the figures as I suggested to them. The authors have included a new section called “Summary” to overview the most relevant conclusions of the review. Overall, I consider that now this review is more solid and it has improved considerably for its publication. For this reason, I encourage to editor to consider it for publication for the interesting value of the review realized, that now it is a much more robust study.